

# Within-host competition drives energy allocation trade-offs in an insect parasitoid

J. Keaton Wilson[1], Laura Ruiz[2] and Goggy Davidowitz[1]

[1] Department of Entomology, University of Arizona, Tucson, AZ, USA
[2] Department of Neuroscience, University of Arizona, Tucson, AZ, USA

## ABSTRACT

Organismal body size is an important biological trait that has broad impacts across scales of biological organization, from cells to ecosystems. Size is also deeply embedded in life history theory, as the size of an individual is one factor that governs the amount of available resources an individual is able to allocate to different structures and systems. A large body of work examining resource allocation across body sizes (allometry) has demonstrated patterns of allocation to different organismal systems and morphologies, and extrapolated rules governing biological structure and organization. However, the full scope of evolutionary and ecological ramifications of these patterns have yet to be realized. Here, we show that density-dependent larval competition in a natural population of insect parasitoids (*Drino rhoeo*: Tachinidae) results in a wide range of body sizes (largest flies are more than six times larger (by mass) than the smallest flies). We describe strong patterns of trade-offs between different body structures linked to dispersal and reproduction that point to life history strategies that differ between both males and females and individuals of different sizes. By better understanding the mechanisms that generate natural variation in body size and subsequent effects on the evolution of life history strategies, we gain better insight into the evolutionary and ecological impacts of insect parasitoids in tri-trophic systems.

## INTRODUCTION

Body size is a biological trait that spans over 21 orders of magnitude (*West, Brown & Enquis, 1999*) with important impacts on chemical and physiological processes and inter- and intra-specific interactions in communities and ecosystems. Size can shape physiological processes like metabolism (*West, Brown & Enquist, 1997*) and is the outcome of complex genetic and developmental mechanisms (*Davidowitz, D'Amico & Nijhout, 2003*; *Davidowitz, Nijhout & Roff, 2012*; *Davidowitz, Roff & Nijhout, 2016*). Furthermore, biologists have long examined patterns of body size over large spatial scales (*Blackburn, Gaston & Loder, 1999*; *Mousseau, 1997*), and examined how body size can affect complex community food webs (*Jonsson, Cohen & Carpenter, 2005*). Examining the relative size of different body parts (allometric scaling relationships (*Gould, 1966*)) has become an important analytical tool alongside measurements of body size in diverse research topics including sexual selection (*Emlen & Nijhout, 2000*; *McCullough & Emlen, 2013*),

Corresponding author
J. Keaton Wilson,
keatonwilson@me.com

physiological adaptation (*Lane et al., 2018*), and animal communication (*Templeton, Greene & Davis, 2005*). Together, body size and allometric scaling relationships not only provide a framework for conserved scaling rules that underlie developmental, physiological, and evolutionary processes (*West, Brown & Enquist, 1997*), but provide an opportunity to examine life history trade-offs, particularly in organisms like insects, where different body structures can act as proxies for investment into different life history strategies (reproduction, dispersal, information-acquisition and cognition).

At its core, life history theory seeks to determine the constraints on selection. Why doesn't selection drive all fitness related traits to high levels and how do organisms differ in rules that govern trade-offs (*Stearns, 1989*; *Fairbairn & Roff, 2006*)? Life history trade-offs (defined as life history traits that are negatively associated with each other; *Zera & Harshman, 2001*) have been studied across a wide variety of taxa (*Zera & Harshman, 2001*), including extensive work on insects, particularly *Drosophila* (*Reznick, 1985*) and crickets (*Mole & Zera, 1994*; *Zera & Denno, 1997*), to determine constraints and mechanisms that govern optimal fitness strategies (*Reznick, 1985*). Work in wing polymorphic crickets has demonstrated a strong trade-off between flight capability and fecundity (*Roff, 1986*; *Zera, Potts & Kobus, 1998*), as well as the importance of nutrient acquisition in shaping trade-offs among traits (*Zera & Brink, 2000*). Though there are many clear examples of trade-offs, results are often mixed as some traits demonstrate little or no trade-offs (*Fernández & Reynolds, 2000*; *McCullough & Emlen, 2013*; *Tigreros & Davidowitz, 2019*). While measurements of allometry involve size (mass) of particular structures, these measurements can prove problematic when addressing life history trade-offs, because allocation costs can be masked in low-mass but energy-dense tissues (*Zera & Harshman, 2001*). Energy provides a common currency with which to compare allocation to different tissues because of energetic differences among molecular building-blocks (i.e. an abdomen and thorax of similar weight may not accurately reflect a similar energetic allocation of resources, as the abdomen may contain a high proportion of energetically dense lipids). Here, we use a combination of weight and energy measurements to examine allocation to different body structures in a parasitoid fly (*Drino rhoeo*).

Tachinid flies (Diptera: Tachinidae) are internal parasitoids of arthropods and typically attack larval stages of herbivorous insects (*Stireman, O'Hara & Wood, 2006*). Tachinids are an understudied yet ecologically important group with approximately 10,000 described species that likely have powerful roles in shaping communities of plants and insects (*Stireman, O'Hara & Wood, 2006*; *Wilson & Woods, 2015*). *D. rhoeo* is a good model for examining trade-offs in resource allocation and patterns in body size because it is a gregarious parasitoid (multiple larvae develop together within a single host) with large variation in cohort size (here meaning groups of *D. rhoeo* larvae developing inside a single host, typically 8–50 larvae within a given host (*Wilson & Woods, 2015*)). This variation means that there is likely competition for host resources at high densities, which can lead to naturally-occurring variation in nutrient acquisition available to larvae, resulting in population-level variation in body size and trait size (*Zera & Harshman, 2001*).

*Drino rhoeo* attacks caterpillars of the hawkmoth *Manduca sexta* and *Manduca quinquemaculata* (*Bernays & Woods, 2000*; *Mira & Bernays, 2002*; *Wilson & Woods, 2015*) frequently at our field site in southeastern Arizona (near Portal and the Chiricahua mountains). Female flies target fourth- and fifth-instar caterpillars, laying eggs on the surface of the caterpillar after which fly larvae shortly (within 20 min) emerge and burrow into the caterpillar hemocoel where they grow and develop. Developing tachinid larvae typically completely consume the bodies of their hosts, resulting in host-death. Previous work has demonstrated that *D. rhoeo* can have a strong impact on the growth and development of hosts post-parasitization, affecting growth, weight and feeding habits (*Wilson & Woods, 2015*) which may, in turn, affect the amount of resources available to the parasitoids.

Here, we use data from a natural population of *D. rhoeo* to address three main questions: (1) does larval competition and host quality drive variation in adult parasitoid body size, (2) what are the energy allocation strategies to different parasitoid body structures (heads, abdomens, thoraces, wings and legs) and how do they vary with parasitoid body size and sex and (3) are there allocation trade-offs among body structures that act as proxies for different life history strategies? To our knowledge, this if the first study that examines larval resource competition in tachinids with an emphasis on energy allocation among different body structures relative to size (resource competition has been shown by others: *Allen & Hunt, 2001*; *Welch, 2006*, *Lehmann, 2008*) and to extend these effects to patterns of life history trade-offs.

## METHODS

### Host and parasitoid collection

Thirty-two *M. sexta* larvae in the fourth or fifth (final) instar were collected from the field near Portal, Arizona (~40 km radius) in August of 2017. Past work with *Drino rhoeo* showed that parasitization rate was high (~44%) in the field for *M. sexta* in fourth and fifth-instar stages (*Wilson & Woods, 2015*), so we collected a mixture of individuals that appeared healthy, and some that appeared to have been attacked by parasitoids (e.g., dried gut fluids on the skin, melanized spots where fly larvae burrowed inside the caterpillar (H.A. Woods and J.K. Wilson, 2014, unpublished data)). Larvae were raised together in a large plastic bin and fed cuttings from local *Datura wrightii*, their main host plant. Once larvae began to wander (*Dominick & Truman, 1984*), they were placed in individual plastic cups (13 cm × 12 cm × 14 cm) filled with soil. Larvae were allowed to burrow and begin pupation and were transported back to the University of Arizona in Tucson where they were kept in an experimental greenhouse for the duration of their development.

### Head-capsule width as a measure of host quality

Here, we define host larval quality as the energy potential of a host to developing parasitoid larvae. Larval mass is not a good measure of larval quality, because growth, feeding and development are all affected by parasitism (*Wilson & Woods, 2015*), so we measured head-capsule width of hosts (all fifth-instar) as a proxy to estimate larval quality to

developing flies. We examine the relationship between mass and head-capsule width in more detail in the Results section.

## Fly emergence, weights and calorimetry

Seven of the 32 caterpillars (22%) collected were parasitized by *Drino rhoeo* and had successful fly emergence. Flies were allowed to emerge in the small plastic cups containing individual caterpillars described above. Once initial fly emergence was observed, we waited 48 h to allow all individuals to successfully emerge from the soil, and then placed the cups in a −20 °C freezer. Once flies were frozen, we transferred them to individually-labeled vials and scored sex using a dissecting scope to determine the presence of sexual patches on the ventral portion of the last abdominal segment on males. Files were sectioned into different body structures (head, abdomen, thorax, wings and legs) using a combination of scalpel, probe and forceps under a dissecting scope. Sectioned body structures were moved to small metal containers and placed in a drying oven at ~45 °C for 48 h before being placed back into vials and frozen again at −20 °C until further processing. Dried individual body sections were weighed on a microbalance (Mettler Toledo XS3DU), with legs and wings being weighed together. In total, we collected 104 individual flies from the seven hosts, though not all flies were used in all subsequent analyses because of some loss of body structures during the weighing or bombing process.

We used a Parr 6200 bomb calorimeter to determine energy content of heads, thoraces and abdomens. Because our calorimeter is designed to be used on larger tissue, we generated calibration curves for predicting caloric content based on tissue type and weight to be able to extrapolate to small weights (and respectively small energy content) that are below the threshold of detection for our calorimeter. Individual structures were mixed into different groups ranging from 1 to 10 structures from individual flies per bin (depending on the size of the tissue: more heads were needed to get to measurable weights than abdomens, because heads are smaller than abdomens) and separated by sex. We also varied the number of structures in binned groups to achieve enough variation in weight to generate accurate calibration curves. Binned tissue samples were weighed and placed in a crucible. We added 0.7 mL of mineral oil to samples as a heat spike (to increase the total energy content to a level readable by the machine). This method of measuring small samples is a standard procedure and the Parr calorimeter software automatically accounts for the mineral oil spike.

*Zera & Harshman (2001)* emphasize that to establish a physiological tradeoff among body functions, there is a need to establish that a specific resource (such as a specific lipid, protein or carbohydrate) is used by both functions which requires tracking the common resource in both functions. In this study we are interested in total amount of resources allocated among functions and not a specific resource. The use of energy as the common currency encompasses all resources allocated to a function and allows for identifying physiological allocation tradeoffs among functions. We note that this method cannot measure the cost of building a structure, just the energy content of the resources that are in the structure.

We created tissue-specific calibration curves to generate estimates of the calorimetric content of different tissues based on weight. We focused on the three main body segments (heads, thoraces and abdomens). To generate curves, we fit linear regression models for each tissue type that were forced through the origin. All models had good predictive power ($p < 0.05$ for all models), though the predictive power for estimating the energetic content of heads ($R^2 = 0.58$) was less than for thoraces ($R^2 = 0.90$) and abdomens ($R^2 = 0.91$) because of smaller sample sizes (it takes many more fly-heads to generate the measurable weights than it does thoraces or abdomens) and more variation in measurements at smaller weights. We did not have enough flies to generate separate calibration curves for males and females, so all bins consisted of flies from a single sex, and calibration curves are tissue-specific but not sex specific, which might result in additional variation in extrapolations if there were large differences between the weight-energy correlations of different sexes.

## Head capsule and host tissue calorimetry

To determine the relationship between head-capsule width and the usable energetic content of hosts, we used late fifth-instar *M. sexta* caterpillars from our colony in Tucson, AZ, USA. We measured head-capsule width and then froze caterpillars in a −20 °C freezer. We then thawed and separated caterpillar structures into two groups: skin and head-capsule and internal tissues that would be available to developing parasitoids (this included hemolymph, tracheae, muscle, but not gut tissue or any remaining food). We dried samples for each caterpillar in a drying oven at ~45 °C for a minimum of 48 h. We crushed dried tissue samples and split samples that were too large for our calorimeter into three sub-samples before bombing. We used a bomb calorimeter (Parr 6200—methods described above) with a 600 μL mineral oil spike to measure the energetic content of each sub-sample before combining for further analysis.

## Data analyses

All analyses were performed in R (Version 3.5.0 'Joy in Playing,' www.r-project.org). We used linear mixed effects models (*nlme*) for modeling the relationship between adult fly size, host quality, cohort size and sex and AIC scores for model comparison and selection. Additional packages (*effects*, *piecewiseSEM*) were used to generate population level trend lines and marginal and conditional $R^2$ values. In spite of our limited sample size of hosts ($n = 7$), we included host as a random effect (with random intercepts) to help control for variance in conditions among hosts. Additionally, we used ordinary least squares (OLS) regression for examining allometric relationships between body weight and tissue weight and energetic content and AIC scores for model comparison and selection (Table S1). Though some researchers have advocated the use of reduced-major axis regression, recent work has shown that OLS regression is better suited in many cases, especially those similar to ours where there is comparatively little measurement error (*Kilmer & Rodríguez, 2017*). In comparisons of relative mass and energy allocation, polynomial models were fit where appropriate. All data and code are archived and available on Zenodo (DOI 10.5281/zenodo.3356991).

## RESULTS

### Cohort size, host quality and adult fly weight

On average, adult fly body size did not differ between males and females (males = 8.74 ± 3.58 mg, females = 8.51 ± 3.73 mg; $F_{1, 90} = 0.091$, $p = 0.7641$). On average, adult fly dry weight decreased with increasing cohort size (Fig. 1A), with an average fly from the largest cohort (38 flies) weighing 36% of an average fly from the smallest cohort (two flies). Additionally, adult fly dry weight increased with increasing host quality (head-capsule width; Fig. 1B). Overall, the best model that explained adult fly weight was one that included cohort size and host quality additively (Tables 1 and 2) with host included as a random effect (random intercepts). We also performed a multiple linear regression on a reduced number of data points ($n = 7$) that were average fly-weight values for each cohort, to confirm the biological pattern we show here and reaffirm that the linear-mixed effects model framework is accounting for any pseudo-replication of sampling multiple flies within a single host. This analysis showed that similar significant effects of cohort size ($p = 0.05$) and host head capsule width ($p = 0.03$) with good predictive power ($p = 0.006$, $R^2 = 0.76$). Head-capsule width is a frequently-used proxy for body size, and typically has strong positive correlations with body size (Smock, 1980; Potter, Davidowitz & Woods, 2011; D'Amico, Davidowitz & Nijhout, 2001)—it is also the best measurement of host-quality in this system because of the complex interactions between parasitism, host-feeding and body size (Wilson & Woods, 2015). We found some support that head-capsule width is positively associated with wandering-weight ($F_{1, 5} = 6.961$, $p = 0.046$, $R^2 = 0.498$) in *M. sexta* caterpillars in the field, despite a relatively small sample size and that caterpillars were parasitized by varying numbers of tachinid larvae. Additionally, in a series of measurements on lab-reared *M. sexta*, we found that pre-wander weight was positively correlated with the caloric content of hosts excluding skin and the gut ($F_{1, 13} = 54.01$, $p < 0.0001$, $R^2 = 0.79$). Together, these data suggest that head-capsule width functions as a good proxy for both body size and caloric content available to developing tachinids.

### Size-relative allocation to different body structures

Overall, flies had relatively smaller heads as size increased, with no difference between males and females ($t = 1.102$, $p = 0.273$), though the best fit model for these data was a second-order polynomial, with a slight increase in investment in heads at large body sizes ($F_{2, 89} = 51.8$, $p < 0.001$, $R^2 = 0.53$; Fig. 2A; Table S1). Normalized wing weight (scaled to individual body size—we use this definition of normalized throughout) decreased with body size ($F_{2, 89} = 10.74$, $p = 0.001$, $R^2 = 0.09$), with no differences between the sexes ($t = -0.768$, $p = 0.444$; Fig. 2B). Normalized leg weight decreased linearly with body size ($F_{2, 89} = 5.169$, $p = 0.008$, $R^2 = 0.50$), and while there was a significant effect of sex ($t = -2.617$, $p = 0.01$), the effect size was small, with little difference between males and females, except only at small body sizes (Fig. 2C). Normalized thorax weight was best fit with a polynomial model that included sex as an additive effect ($F_{3, 88} = 35.98$, $p < 0.001$, $R^2 = 0.54$)—relative thorax weights were highest in medium-sized flies, and were higher
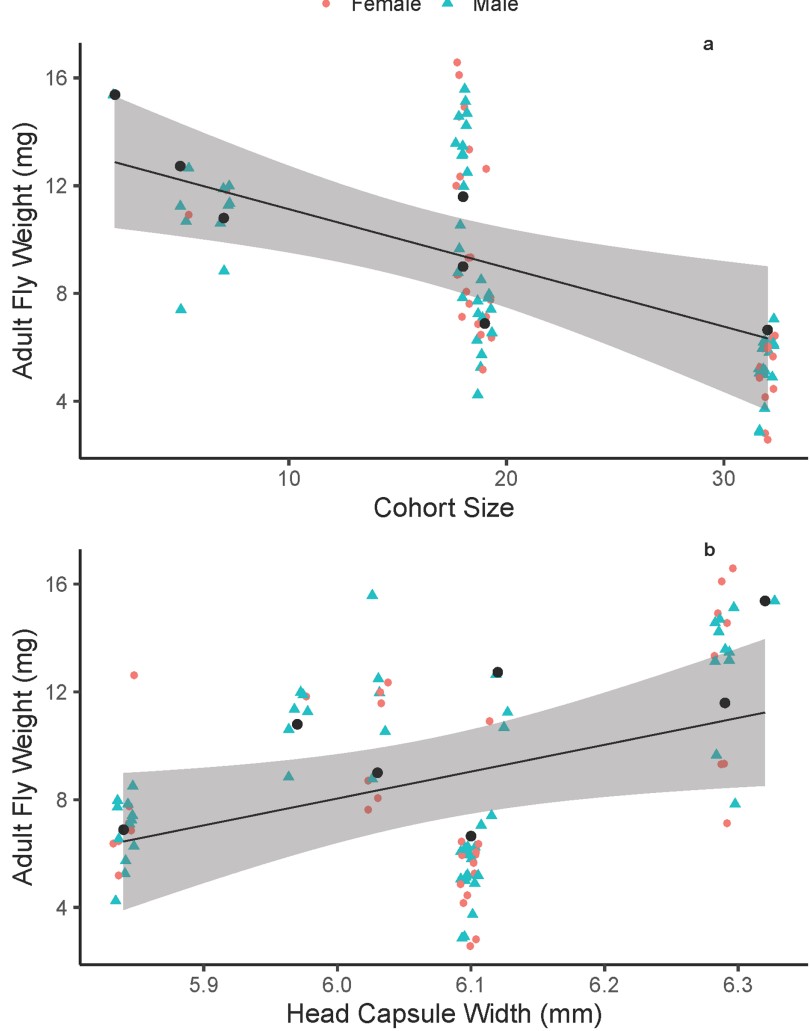

**Figure 1** **Adult *Drino rhoeo* weight as a function of competitive larval environment (cohort size) and host quality.** (A) The negative relationship between increasing larval competition and adult size. (B) The positive relationship between host quality (head-capsule width) and adult fly weight. In both panels, green points represent male flies and red points female flies. Black lines represent the trendline for the best fit linear mixed effects model that included cohort size and head-capsule width as fixed effects and host as a random effect. Gray bands represent the 95% confidence interval. Raw data is jittered horizontally slightly to aid in visualization, and larger black points represent the predicted adult fly weight for each host.                                    

overall in males ($t = 7.909$, $p < 0.001$), with the greatest difference between males and females occurring in medium-sized flies (Fig. 2D). Finally, normalized abdomen weight increased linearly with fly weight in both males and females ($F_{2, 89} = 65.18$, $p < 0.001$, $R^2 = 0.56$), though females showed increased relative investment across all body weights ($t = -8.399$, $p < 0.001$; Fig. 2E).

## Energy tradeoffs and comparisons among body segments

Body segments differed in their average energy content with abdomens being the most energy dense at 4.36 ± 0.44 calories/mg, followed by thoraces at 4.05 ± 0.33 calories/mg,

**Table 1 Model selection for factors affecting adult fly weight.**

| Model | Marginal $R^2$ | Conditional $R^2$ | AIC |
|---|---|---|---|
| Weight ~ cohort size | 0.338 | 0.716 | 408 |
| **Weight ~ cohort size + head-capsule width** | **0.462** | **0.683** | **401** |
| Weight ~ cohort size * head-capsule width | 0.446 | 0.706 | 402 |
| Weight ~ cohort size * head-capsule width * Sex | 0.438 | 0.699 | 407 |

Note:
Bold indicates the final model used in further analysis and data visualization.

**Table 2 Linear mixed effects model predicting fly weight as a function of cohort size and head-capsule width.**

| Fixed Effects | | | | $t$-Value | $p$-Value |
|---|---|---|---|---|---|
| Parameter | Estimate | Standard error | d.f. | | |
| Intercept | −47.205 | 0.1909 | 85 | −1.723 | 0.09 |
| Cohort size | −0.218 | 0.2727 | 4 | −3.000 | 0.04 |
| Head Capsule Width | 9.972 | 0.2466 | 4 | 2.226 | 0.09 |
| **Random effects (host–random intercepts)** | | | | | |
| Parameter | Intercept | Residual | | | |
| StdDev | 0.1947 | 1.935 | | | |

followed by heads at 1.96 ± 0.17 calories/mg. We compared the percentage of calories devoted to each body segment for male and female flies across body sizes and found strong potential trade-offs between thoraces and abdomens in both males and females, though the pattern is strongest for male flies of moderate size (Fig. 3A). We note here that while a negative correlation between two traits has often been used as evidence for trade-offs (*Zera & Harshman, 2001*; *Stearns, 1989*), the interactions among traits are complex, and indirect effects may impact negative correlations. We also compared the relative amount of energy devoted to one segment after controlling for the total energy content of the three main body segments. Both male and female flies showed a strong negative correlation between the relative amount of energy invested in thoraces and abdomens ($F_{1, 99} = 1983$, $p < 0.001$, $R^2 = 0.952$; Fig. 3B), with no difference between males and females ($p = 0.209$). There was no significant pattern of trade-offs between heads and thoraces or heads and abdomens for male or female flies ($F_{1, 99} = 1.225$, $p = 0.271$; $F_{1, 99} = 1.187$, $p = 0.279$).

## DISCUSSION

*Drino rhoeo* show clear indications of strong density-dependent larval competition (Fig. 2), similar to other gregarious insect parasitoids (*Taylor, 1988*; *Harvey, 2000*; *Allen & Hunt, 2001*). This competition (in combination with variation in host quality; Fig. 2B; Table 1), leads to a wide range of body sizes that is naturally occurring, and is not sexually dimorphic (Fig. 1). In spite of a low sample size of hosts ($n = 7$), we found strong relationships between cohort size and adult fly weight, as well as moderate impacts of

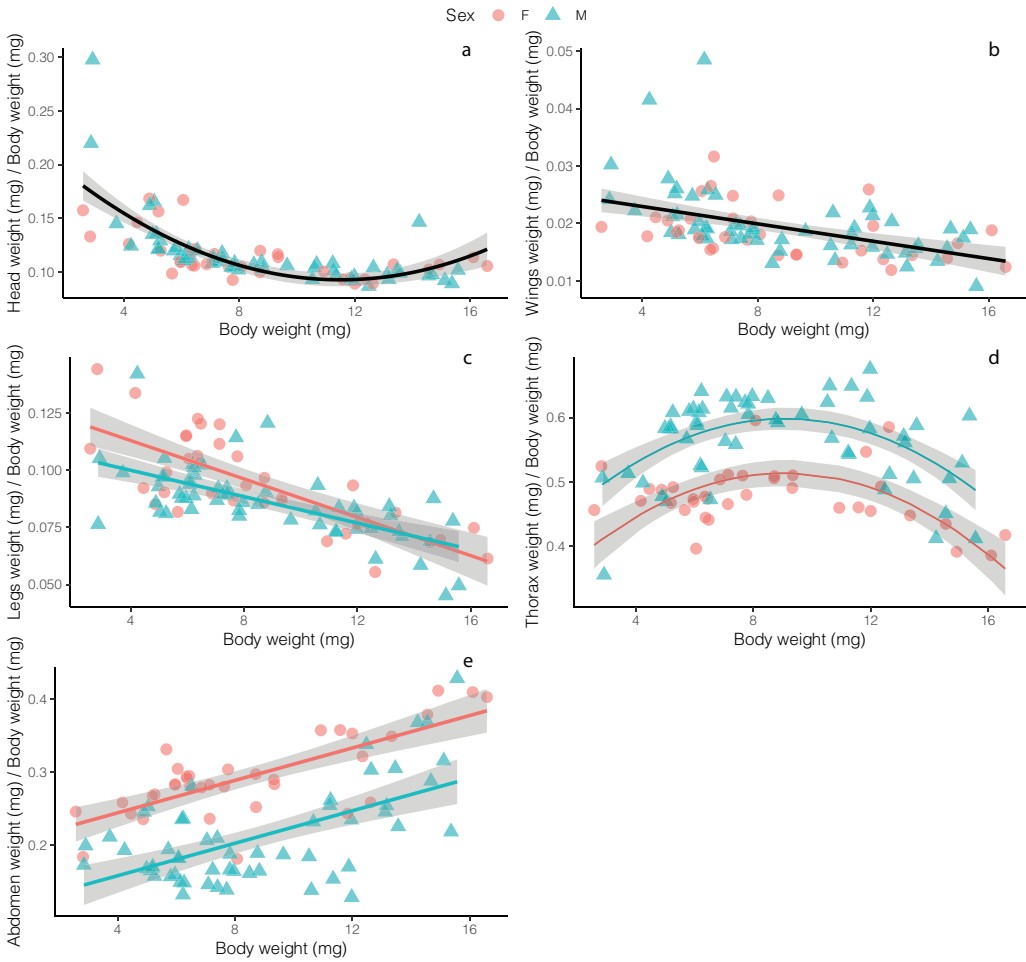

**Figure 2 Relative weight investment in different adult *Drino rhoeo* body structures as a function of body size.** Different panels show relationships of relative weight investment of (A) heads, (B) wings, (C) legs, (D) thoraces and (E) abdomens. Males flies are represented by green dots whereas females are represented by red dots. Solid lines depict the trendlines for the best fit model for each body structure whereas gray bands represent the 95% confidence interval. In panels with two lines depicted (D and E), there was a significant difference between males and females.

host quality on adult fly weight. The evolutionary and ecological implications of density-dependent larval competition in insect parasitoids have been examined only in hymenopteran parasitoids previously (*Nicol & Mackauer, 1999*; *Milonas, 2005*; *Sykes et al., 2007*). Our results presented here pave the way for further work examining tradeoffs between immune function and larval competition, female tachinid oviposition strategies and optimal brood size in non-hymenopteran parasitoids.

Allometric scaling relationships of the mass of tissues provide insight into how investment in structures change across body sizes and sexes. Flies invested in relatively larger wings, legs, and heads at smaller body sizes while investing in relatively larger abdomens at high body sizes and large thoraces at moderate body sizes (Fig. 2). Male and female flies differed markedly in their allocation to thoraces and abdomens with males

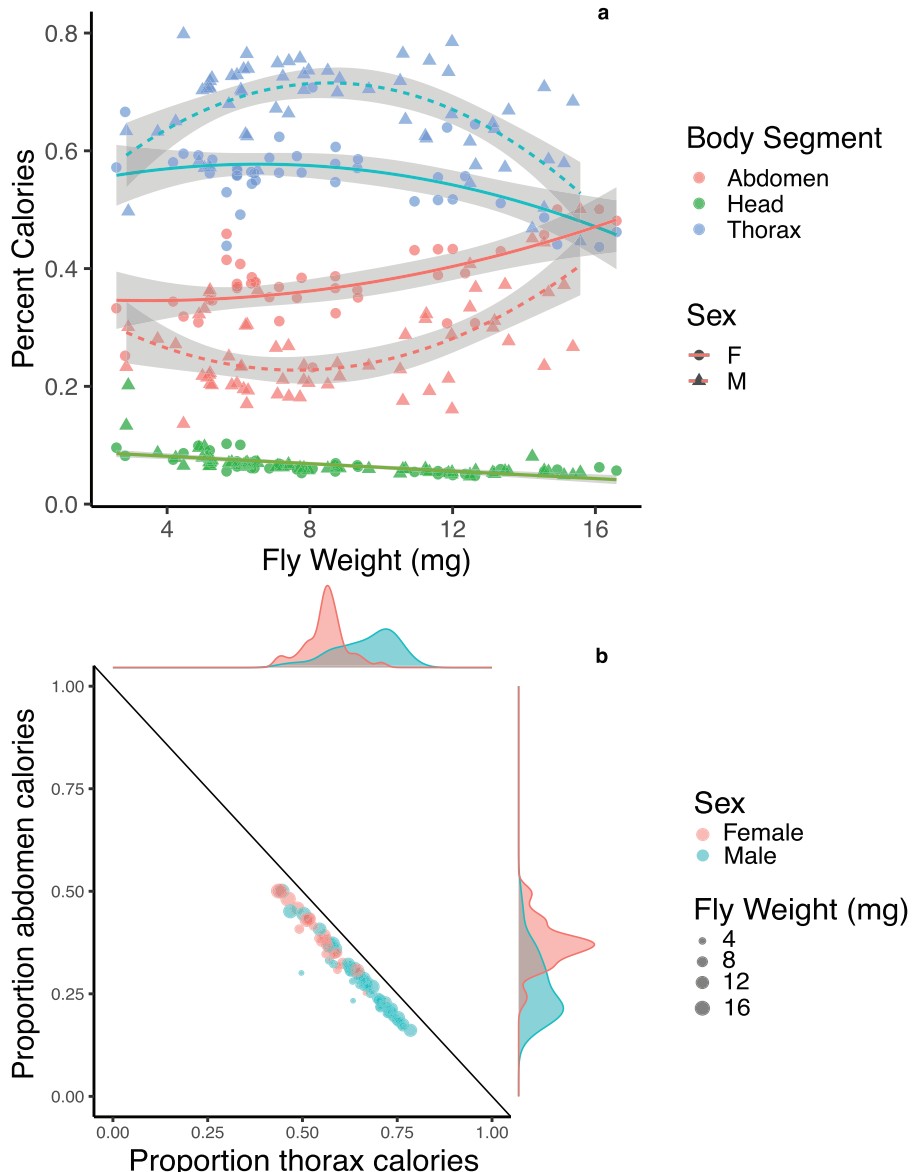

**Figure 3 Energetic trade-offs in allocation to different body segments of adult *Drino rhoeo*.** (A) The percent calories of heads, abdomens and thoraces for male and female flies as a function of body size. Abdomens (red) and thoraces (blue) show significant differences between males (triangles) and females (circles), while there was no difference between males and females in energy allocation to heads (green). (B) Energy allocation trade-off between abdomens and thoraces for *Drino rhoeo*. Individual abdomen and thorax energy content is normalized for total energetic content of all three main body segments and plotted against each other. Female flies are in red and male flies are in green, while body size is represented by the size of the circle. Marginal density plots depict the differences in the allocation of resources to thoraces and abdomens between males and females.

investing relatively more in thoraces and females investing relatively more in abdomens (Figs. 2D and 2E). Fly legs, wings and heads showed hypoallometric scaling relationships (scaling coefficients of 0.709, 0.559, 0.68, respectively; Fig. S1), while thoraces for both males and female flies were isometric (scaling coefficient of 0.986). Abdomens for both

males and females were hyperallometric (scaling coefficient of 1.236), mirroring previous work in other insects (*Wickman & Karlsson, 1989*).

Both male and female flies showed strong trade-offs between abdomens and thoraces in both mass (Fig. 2) and energetic content (Fig. 3B). Though this trade-off is consistent across body sizes, relative allocation to each structure is markedly different between males and females and across body sizes (Fig. 3A). As a group, the relative investment into abdomens or thoraces falls along a clear gradient—with females devoting energy to abdomens at the expense of thoraces, while the opposite is true for males (Fig. 3B). Though the majority of flies follow this allocation rule, a few small male flies in our study demonstrate a different allocation strategy—less investment overall to abdomens and thoraces while devoting more energy to the head (Fig. 3B). Investment in thoraces (a proxy for dispersal, though many insects are capable of resorbing wing musculature as adults (*Stjernholm, Karlsson & Boggs, 2005*; *Boggs, 2009*)) is favored at small and moderate body sizes, while equal investment in thoraces and abdomens (reproduction and storage) is favored at large body sizes. These patterns are magnified in male flies, where some individuals invest up to 80% of the total energetic content of the three main body segments into their thoraces, and dampened in females. There is evidence in other insect systems that increased allocation to thoraces is correlated with higher flight performance (*Berwaerts, Van Dyck & Aerts, 2002*; *Karlsson & Johansson, 2008*), though the energy allocation to thoraces we present here may also represent a minimum threshold required to produce functional flight. Researchers have also shown that increased allocation to abdomens is correlated with increased fecundity (*Wickman & Karlsson, 1989*; *Griffith, 1994*; *Preziosi et al., 1996*). Here, flies demonstrate strategies that closely match predictions made by life history theory for income-breeding insects (*Davis et al., 2016*) where dispersal and mobility are favored in poor quality environments (tachinids are likely income breeders that feed on nectar and pollen (*Gilbert & Jervis, 1998*; *Tooker, Hauser & Hanks, 2006*) and can ameliorate larval nutritional deficits as adults by being able to find high-quality resource sites) while a bigger relative investment in reproduction or energetic storage can be allocated in high quality larval environments (*Boggs, 2009*). These allocation patterns are likely magnified in males and dampened in females because of heightened energy requirements for reproductive investment and oviposition success (*Reznick, 1985*).

## CONCLUSIONS

In conclusion, we show that larval competition and variation in host quality generate a range of body sizes in a population of insect parasitoids which in turn leads to different patterns of allocation to thoraces and abdomens across body sizes and between sexes. These strategies point to strong trade-offs between body segments tied to reproduction and dispersal. By better understanding the mechanisms that drive allocation to different tissues across body sizes and between sexes, particularly in under-studied species with potentially large ecological effects, we gain deeper insight into the evolution and ecology of tri-trophic systems and the underlying drivers of life history strategies.

## ACKNOWLEDGEMENTS

We'd like to thank the director of the Southwestern Research Station and station employes for their help, as well as Cristina Francois, Lennie Park, Natasha Tigreros and Meck Slagle for their input on these data, analyses and presentation. We'd also like to thank Heather Costa for her support in the lab and for input on data, analyses and presentation.

### Funding

This study was supported by National Science Foundation grant IOS-1053318 to Dr. Goggy Davidowitz and by the Center for Insect Science National Institutes of Health Postdoctoral Excellence in Research and Teaching (PERT) grant K12GM000708. There was no additional external funding received for this study. The funders had no role in study design, data collection and analysis, decision to publish, or preparation of the manuscript.

### Grant Disclosures

The following grant information was disclosed by the authors:
National Science Foundation: IOS-1053318.
Center for Insect Science National Institutes of Health Postdoctoral Excellence in Research and Teaching (PERT): K12GM000708.

### Competing Interests

The authors declare that they have no competing interests.

### Author Contributions

- J. Keaton Wilson conceived and designed the experiments, performed the experiments, analyzed the data, prepared figures and/or tables, authored or reviewed drafts of the paper, and approved the final draft.
- Laura Ruiz performed the experiments, analyzed the data, authored or reviewed drafts of the paper, and approved the final draft.
- Goggy Davidowitz conceived and designed the experiments, authored or reviewed drafts of the paper, and approved the final draft.

### Data Availability

All data and code are available at Zenodo: Keaton Wilson. (31 July 2019). keatonwilson/tachinid-competition-development: Initial release for Zenodo archive (Version v1.0). Zenodo. DOI 10.5281/zenodo.3356991.

### Supplemental Information

Supplemental information for this article can be found online at http://dx.doi.org/10.7717/peerj.8810#supplemental-information.

Wilson et al. (2020), *PeerJ*, DOI 10.7717/peerj.8810
12/15

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
