# Peer review of "Within-host competition drives energy allocation trade-offs in an insect parasitoid"

_PeerJ, doi:10.7717/peerj.8810_

## Round 0.1 · original submission · Major Revisions

Both reviewers felt that the study showed sufficient scientific merit for publication; however both reviewers raised substantive issues that would require attention before the manuscript would be ready for publication. Of particular concern is that the manuscript has not included a large body of existing literature (Reviewer 1) and placed the manuscript's findings in this context. Related to this, Reviewer 2's concern that the manuscript needs to be re-framed towards where the manuscript sets-up and tests a particular set of hypotheses based on current theory. This would again place the manuscript's findings in context of what is currently known and highlight any new results. Reviewer 1's also raised concerns related to the statistical treatment of the non-independence of each fly sample. One would need to reanalyze effects of cohort size and host quality on fly size avoiding any possible pseudoreplication and make it clear that individual flies are not independent in other analyses. Each reviewer also raised a number of minor issues which they would like to see addressed in order to improve the manuscript. Please pay particular attention to grammar and use of of standard and full English words.

Reviewer 1 ·

Basic reporting

This manuscript is clearly written for the most part. A few passages are a bit awkward, and some statements, especially in the Introduction seem vague or imprecise and could be improved with careful proofreading. For example:
Abstract Lines 53-56 this is a bit vague. Also, the first “rules” could be omitted.
Introduction
L. 82-consequences for?
L. 85 “evolutionary target” – evolution has no target. I’m not sure what is meant here.
L. 86-87 to what end?
L. 87-88 In what way?
L. 94 – what is meant here by “outline”?
L. 96. Insects are not really modular, like a coral for example. They are metameric, with segments specialized as groups into tagmata. The use of terms like body parts and segments in the manuscript is confusing, as a segment has a specific definition in insects. Perhaps they should use terms such as tagmata and appendages
L. 100 constraints on selection
L. 101 fitness related traits?
L. 104,106 work on
L. 109 tradeoffs between what? Be more precise.
Some citations include first and middle initials, some don’t, some have full first names written out, some are italicized…
L. 111-112 doesn’t allometry always involve size? And generally not tissues but structures or organs?
L. 126 Always spell out generic names at the beginning of a sentence.
L. 136 – the authors can’t mention their field site without saying something about it (e.g., where it is)
L. 171 these are usually referred to as sexual patches (Cerretti et al, 2014 First report of exocrine epithelial glands in oestroid flies: the tachinid sexual patches (Diptera: Oestroidea: Tachinidae). Nuptial pad is a new and unnecessary term.
L. 206 Tracheae
L. 229 It is probably best to avoid contraction like didn’t
There are also punctuation errors, e.g., extra ) or . that need to be fixed

The authors provide a brief overview of life history theory and allometry that is sufficient, though see comments above on precision. On line 146 they state that theirs is the first study to examine cohort resource competition in Tachinidae, but this is false. In fact, many studies have examined numbers and size and survival of tachinids in hosts observationally and experimentally. This study fails consider this previous literature and place its results in this context, and this is a major fault. However, this is likely the first study to assess energetic allocation among tagamata relative to size.
Structure - fine
Figures – The figures are generally well-constructed and illustrate the man results. Perhaps make males and females easier to distinguish if the paper is printed in grayscale.
Data from the paper appear to be available on Zenodo

Experimental design

The study fall within the scope of the journal
The questions are clearly stated
The authors use each fly as an independent sample in analyses, yet they come from only 7 hosts. This does not seem appropriate. Flies within a host experience the conditions of that particular hosts and interact with one another. In addition, they are likely to be genetic siblings. Thus it seems that mean values per host should be used as independent data points. The authors include host as a random variable to account for this non-independence, but this still strikes me as pseudoreplication. The patterns may remain with N=7, but they may be altered and significance of some factors may change.
For assessment of energy allocation, it may be more appropriate to treat each fly as an independent sample despite genetic non-independence.
Is the energetic content of lab-reared hosts relative to head widths the same as those in the field?
L. 183 it is a little unclear what is meant by “binned?” How many measures were taken per body region?
What is the total number of flies examined?
L. 213 Joy in playing???

Validity of the findings

The study fall within the scope of the journal
The questions are clearly stated
The authors use each fly as an independent sample in analyses, yet they come from only 7 hosts. This does not seem appropriate. Flies within a host experience the conditions of that particular hosts and interact with one another. In addition, they are likely to be genetic siblings. Thus it seems that mean values per host should be used as independent data points. The authors include host as a random variable to account for this non-independence, but this still strikes me as pseudoreplication. The patterns may remain with N=7, but they may be altered and significance of some factors may change.
For assessment of energy allocation, it may be more appropriate to treat each fly as an independent sample despite genetic non-independence.
Is the energetic content of lab-reared hosts relative to head widths the same as those in the field?
L. 183 it is a little unclear what is meant by “binned?” How many measures were taken per body region?
What is the total number of flies examined?
L. 213 Joy in playing???

Additional comments

see other fields

Reviewer 2 ·

Basic reporting

OK

Experimental design

The aims are a bit too desciptive and therefore of not too high relevance. Some rewriting can help.

Validity of the findings

Statitical treatment should be more transparent in some aspects.

Additional comments

This study shows that parameters of the host influence body sizes of a gregarious dipteran parasitoid, and that flies of different sizes make different allocation decisions. There seems to be nothing fatally wrong with the study, and the piece of knowledge provided by this paper is definitely welcome. Nevertheless, everything remains at a quite descriptive level, the choice of the examined relationships appears not to be based of any theory-driven a priori hypotheses, and, consistently, the results appear to be underdiscussed. The reader is left with an “and so what?” –feeling. Should we have expected something different, and why?

Specific/minor:
• It would add to information content of the Abstract if you said something about the magnitude of the differences, e.g. the flies differed in weight X.X times.
• Line 59. Any conclusions about reproduction and dispersal are indirect at best, so these should not be mentioned in the Abstract at least.
• Tell us more about the biology of the tachinid. There are aspects important in the context of the present study. Do the parasitoids fully consume the bodies of their hosts? (If not, we should not expect much competition). Do the females adjust their clutch sizes to the size of the host? (If yes, this could eliminate the effect of host size on competition intensity).
• Line 81, 103 and in many other places. Given names are not normally included in such references.
• Second paragraph of the Introduction. I think that here the authors should explain how do they understand/define life-history trade-offs in the context of this study. I think that this would contribute to clarifying both the authors’ and the readers’ thoughts (see below about inferring about trade-offs).
• Line 125. Also of plants?
• The description of methods is too detailed in parts. For example, it does not help if we know the size of the cups (line 159), neither need we know that the insects were transported back to Tucson (line 160). Details which could not have affected the results in any way should not be presented.
• Line 162. It is not very clear what do you mean with ‘quality’ of the host larvae. In particular, do you imply that that there are aspects of quality other than size? If yes, please specify. If not, I think that it is better to avoid the ambiguous term ‘quality’ and to talk about size only.
• Tell us more about head capsule width. I guess that you mean that head capsule width of last instar describes the weight of the larvae in the beginning of the last instar, and tells us how much food the parasitoids actually had, while the mass of the larva later during the development is affected by parasitism and is therefore not a reliable measure of the initial amount of the resources. If so, please say this more directly.
• Line 194. I do not understand ‘robust’ here, may be my fault.
• Line 198-200. I do not fully understand this. Do you mean that you derived a size vs energy content relationship which was common to both sexes? If so, I think that this is not good because the content of abdomen is very different in males and females. If not, please reformulate to avoid misunderstandings.
• Data analyses. OK, you nicely tell us which models did you have for the analyses of size but you do not tell in sufficient detail how did you select the models to drive the allometric relationships which still constitute the primary output of the paper. How was host individual treated in those analyses? I think that these models should be presented in full in a supplement as a minimum.
• Line 237. Body size is a vague term. Perhaps ‘with body mass’?
• Lines 238-239. I do not understand.
• Line 252. “Normalized” can mean different things. Be more specific.
• Line 268 and elsewhere: most definitely, you cannot conclude about the existence of trade-offs on the basis of just correlative data. A negative correlation between two things can arise in an indirect way, a correlation as such does not prove a trade-off. As a maximum, you can say that you found a pattern consistent with a potential trade-off.
• Line 285. “only in hymenopterous…” may be misleading. There are lots of such studies on Drosophila I guess?
• Line 313. “Flies …”. Here starts a superlong sentence which must be reformulated for the sake of clarity.
• Line 315. I would cite Davis, R. B., Javoiš, J., Kaasik, A., Õunap, E. and Tammaru, T. 2016. An ordination of life-histories using morphological proxies: capital vs income breeding in insects. - Ecology , 97: 2112-2124 instead of Tammaru and Haukioja here.
• References carelessly prepared, e.g. given names of the authors are sometimes presented (unusual), sometimes not.
• Figure 1 text. “predicted” little unclear here. Averages corrected for sex?
• May be this is old-fashioned but I am used to the requirement that figures should be readable also in black and white. Your “green” and “red” cannot be distinguished in my printout.
• Figure 2. Is “fly weight” (x-axis) and “body weight” (y-axis) the same thing? If yes, please unify. If not, please explain. If yes, then may there be a problem: the variables on the axes contain common elements which may lead to artefacts: Brett, M. T. 2004. When is a correlation between non- independent variables “spurious”? Oikos 105:647–656. Please check and comment.
• I understand that I should not teach native speakers but I am not used to see “didn’t” etc in scholarly texts.

---

## Round 0.2 · Major Revisions

I very much appreciate the time and effort you have undertaken to address the first round of Reviewers' comments and suggested edits. This has certainly improved the manuscript.

Reviewer 2 has reviewed your revised manuscript, and although highlighting the scientific merit of the study, has raised a few issues that they feel require addressing/explaining before the manuscript can be considered for publication. In particular, their points 2-4.

It is important to obtain clarity on these more substantive points raised by Reviewer 2, as they will possibly require adjustments to your methods, results and figures. Hence, the decision of "major revision".

Reviewer 2 ·

Basic reporting

OK.

Experimental design

OK.

Validity of the findings

Still some problems with transparency of the statistical treatment. I disagree with some of the interpretations.

Additional comments

I still think that this study has a merit, and I also see various improvements. Nevertheless, the following problems remain.

1. Still, are there any other studies on insects which show how allocation of resources to different parts of body depend on body size? Hard to believe that there aren’t any. If there is something to compare to, please do so explicitly.

2. I still feel uncomfortable with your conclusions about trade-offs in resource allocation. I hope that I am now able to formulate my feelings more clearly. You seem to base your conclusions about the trade-off on what we see in Fig 3b. I am sorry but I would say that this relationship is quite trivial. Let’s consider an analogous example: assume that we cut loafs of bread into two parts (not necessarily equal) and then we study the relationship between relative weights of the left part and the right part. We will get a straight line with the slope equal to -1. We may call this a reflection of a trade-off between the weights of the right parts and the left parts if we wish but, in fact, this is nothing more than a mathematical triviality. Your Fig 3b is not much different: there is some little random noise around the line due to the fact that there is also some allocation to the rest of the body but as thorax and abdomen constitute by far the largest segments of the body, the line is still close to b=-1 with little residual variance. This is a mathematical inevitability and not a biologically meaningful discovery of a strong trade-off. So I suggest that you remove Fig 3b and all the discussion based on it (Figures 2 and 3a are informative and valuable). If you disagree then please show us how would it be possible to get a different result in this case, i.e. a pattern which you would interpret as no trade-off, or as a weak trade-off.

3. I rather disagree with your interpretation that the higher relative investment into thoraces in small flies is an adaption to make it possible to escape adverse conditions (line 347). I think that the thing is more simple. I think that it is generally valid for insects and perhaps any organisms that movement-related structures show negative allometries (show less relative variation than the size of the entire body) just because for any given body plan, these structures cannot vary much in size and remain functionality at the same time (if you increase linear measures of a bird 2 times maintaining its body shape it will not be able to fly). That’s why thorax size is more conserved than abdomen size: the thorax is built of the size allowing the insect to fly, and all the rest is allocated into the abdomen. The amount of ‘the rest’ may vary substantially as a function of environmental conditions.

4. You seem to disagree with my criticism of your statistical treatment not being fully transparent. However, for example, in line 277 you say that the best fit was an 2nd-order polynomian: where is this analysis explained? Which functions were compared and how when you decided which non-linear functions best describe the relationships in Figs 2 and 3a? Where are the details?

Minor/specific

line 59. I still insist that ‘body size’ is a too vague term to be used in quantitative statements. Were the flies 6 times larger in terms of body masses or linear measurements of the body? A 6-fold increase in linear measurements corresponds to a 216-fold increase in body mass, given that body proportions do not change.

line 61. I still think that there is insufficient evidence for different life history strategies. I do not
see anything which could not be explained by biomechanically based allometric rules (see above).

line 101. isn’t ‘rules that govern trade-offs’ an oversophisticated expression? Would you be able to give an example of such a rule?

line 119-120. I feel that the sentence ‘Here, …’ is in a wrong place. Does not fit here.

Line 125. Once again: do tachinids really shape plant communities? They are all parasitoids as far as I know.

Line 128. I think that you should explain how do you use the term ’cohort’ here. In population biology, this word has a different meaning (= individuals which were born during a certain period of time), and this may cause confusion.

Line 166. Thank you for explaining the usage of ‘quality’ here, this is essential indeed. However, I feel that this text (starting with ‘Here, we…’) does not fit well here as it stands. As this is in the Methods section, maybe you could start with telling that you measured head capsulae, and then you could smoothly switch to explaining why did you do this. I think that this deserves a separate paragraph if not a separate section. Also, you have somewhat repetitive text starting line 263, may it be possible to combine these texts to avoid repetition?

Line 202. I think that this paragraph is not needed.

Line 220. in there -> if there?

Line 237. I still do not understand ‘population level’ here? Which population?

Once again, please note that in your references list, you sometimes present given names and sometimes not. I did not check it but this is hardly a style acceptable by PeerJ or any journal.

Figures. It is great to know that most readers will read your paper online but, nevertheless, please change your colour scheme so that there would be a slight difference also in the darkness of different symbols, not only in the wavelength. This is easy to do and would not hurt your online readers in any way. Please.

Figure 2. Once again, is ‘fly weight’ (horizontal axis) and ‘body weight’ (vertical axis) the same thing or not? If yes, please use identical terms on both axes. If not, please explain the difference.
* * *

---

## Round 0.3 · Minor Revisions

Thank you for attending to Reviewer 2's main comments and for the detailed explanation concerning comment #2.

I would like to ask one final request before acceptance. You have not dealt with all of Reviewer 2's specific comments. Some of these were perhaps more differences of opinion and you preferred not to change these, but I ask you to take a second look at all of then, particularly the ones pasted below, which I agree with the Reviewer, and which require dealing with, and please respond to the comments.

This may seem somewhat pedantic, but having you deal with these last few minor comments now will speed the progress of your manuscript at the next stage.

"Line 166. ...I feel that this text (starting with ‘Here, we…’) does not fit well here as it stands. As this is in the Methods section, maybe you could start with telling that you measured head capsulae, and then you could smoothly switch to explaining why did you do this. I think that this deserves a separate paragraph if not a separate section. Also, you have somewhat repetitive text starting line 263, may it be possible to combine these texts to avoid repetition?"

"Line 220. in there -> if there?"

---

## Round 0.4 · accepted · Accept

Thank you for attending to these last few minor comments - your manuscript has now been accepted.

I noticed a few double spaces have crept into the text, such as before a bracket. When you do a final proofread, keep an eye out for these.

Once again, thank you for all you efforts dealing with Reviewer comments and editing your manuscript.